# Sarcopenia as a Prognostic Marker in Elderly Head and Neck Squamous Cell Carcinoma Patients Undergoing (Chemo-)Radiation

**DOI:** 10.3390/cancers14225536

**Published:** 2022-11-10

**Authors:** Erik Haehl, Luisa Alvino, Alexander Rühle, Jiadai Zou, Alexander Fabian, Anca-Ligia Grosu, Nils H. Nicolay

**Affiliations:** 1Department of Radiation Oncology, University of Freiburg—Medical Center, Robert-Koch-Str. 3, 79106 Freiburg, Germany; 2German Cancer Consortium (DKTK) Partner Site Freiburg, German Cancer Research Center (dkfz), Neuenheimer Feld 280, 69120 Heidelberg, Germany; 3Department of Radiation Oncology, University Hospital, LMU Munich, 81377 Munich, Germany; 4Department of Radiation Oncology, University of Leipzig Medical Center, Stephanstr. 9a, 04103 Leipzig, Germany

**Keywords:** sarcopenia, head-and-neck cancer, head-and-neck squamous cell carcinoma, radiotherapy, elderly, HNSCC

## Abstract

**Simple Summary:**

Sarcopenia, i.e., the loss of muscle mass is a risk factor for reduced survival and increased treatment-associated toxicities in many cancers, and it increases with age. In this analysis, we investigated sarcopenia and sarcopenia dynamics during treatment in elderly patients (>65 yrs) with head-and-neck cancers undergoing (chemo)radiotherapy. Patients who exhibited sarcopenia prior to radiotherapy had a significantly reduced overall survival as well as an increased risk for incurring higher-grade toxicities. Additionally, pre-treatment sarcopenia in elderly patients correlated with a decreased body weight, a higher level of comorbidities, larger tumor size and increasing age. During (chemo)radiotherapy, muscle mass in patients remained relatively stable, and post-therapeutic sarcopenia did no longer correlate with patient survival. Considering the vulnerability of elderly patients with head-and-neck cancers, assessing sarcopenia prior to radiotherapy may help in shared decision-making regarding the optimal treatment. Additionally, sarcopenia can be a valuable predictive marker regarding the necessity of early supportive measures in elderly patients with head-and-neck cancers undergoing radiotherapy.

**Abstract:**

Sarcopenia is associated with reduced survival and increased toxicity in malignant diseases. The prevalence of sarcopenia increases with age and is an important cause of functional decline. We analyzed sarcopenia and sarcopenia dynamics in elderly head-and-neck squamous cell carcinoma (HNSCC) patients undergoing (chemo)radiation. Skeletal muscle mass of 280 elderly HNSCC-patients (>65 yrs) receiving curative (chemo)radiation was manually outlined and quantified on CT scans at the level of the C3 (C3MA). Cross-sectional muscle area at L3 (L3MA) was calculated and normalized to height (L3MI). Frequency distributions of clinical parameters as well as overall survival (OS), progression-free survival (PFS) and locoregional control (LRC) were calculated regarding sarcopenia. Calculated L3MA correlated with pretherapeutic hemoglobin-levels (ρ = 0.280) bodyweight (ρ = 0.702) and inversely with patient-age (ρ = −0.290). Sarcopenic patients featured larger tumors (T3/4 69.0% vs. 52.8%, *p* < 0.001), a higher burden of comorbidity (age-adjusted Charlson Comorbidity Index 4.8 vs. 4.2, *p* = 0.015) and more severe chronic toxicities (CTCAE grade 3/4 24.0% vs. 11.8%, *p* = 0.022). OS was significantly deteriorated in sarcopenic patients with a median of 23 vs. 91 months (logrank *p* = 0.002) (HR 1.79, CI 1.22–2.60, *p* = 0.003) and sarcopenia remained an independent prognostic factor for reduced OS in the multivariate analysis (HR 1.64, CI 1.07–2.52, *p* = 0.023). After therapy, 33% of previously non-sarcopenic patients developed sarcopenia, while 97% of pre-treatment sarcopenic remained sarcopenic. Median bodyweight decreased by 6.8%, whereas median calculated L3MA decreased by 2.4%. In contrast to pretherapeutic, post-therapeutic sarcopenia is no prognosticator for reduced OS. Pretherapeutic sarcopenia is a significant prognostic factor in elderly HNSCC patients undergoing (chemo-)radiation and should be considered in pretherapeutic decision-making. Its role as a predictive marker for tailored supportive interventions merits further prospective evaluation.

## 1. Introduction

Head and neck squamous cell carcinoma (HNSCC) is a common malignancy associated with alcohol and tobacco intake and human papillomavirus infection [1,2]. Primary therapy comprises surgery or radiotherapy for early stages and multimodal therapy for advanced stages [3]. Although cure can be achieved even in locally advanced cancers, therapy holds significant acute and late toxicities [4].

Sarcopenia is a skeletal muscle disorder that is defined by reduced muscle function and low skeletal muscle mass. It is primarily prevalent in older adults due to age-associated muscle-loss, but also caused by additional factors including malnutrition, inactivity, neurological disorders and malignant neoplasms [5]. In oncology, progressive sarcopenia is part of a multifactorial syndrome referred to as cancer cachexia that with or without a loss of fat mass leads to functional decline and cannot be fully reversed by nutritional support [6]. Both are symptoms of the oncologic disease and negative prognostic factors in many malignancies [7].

Sarcopenia, derived from CT-based skeletal muscle quantification, has been shown to correlate with adverse outcomes in various cancer sites, such as lung, bladder or rectal cancers [8,9,10]. Usually skeletal muscle area (SMA) at a certain level is obtained from CT scans and normalized to body height to receive the skeletal muscle index (SMI). SMI has been shown to correlate closely with the gold standard of whole-body muscle mass measurements [11].

Head-and-neck cancer patients are especially prone to sarcopenia due to disease- and treatment-related malnutrition and dysphagia [12]. Accordingly, two recent reviews identified CT-based sarcopenia as a negative predictor of overall survival (OS) in HNSCC patients undergoing curative (chemo)radiation [13,14]. For the head-and-neck region, cross sectional muscle mass is mostly obtained from the level of the third cervical vertebra, which showed a very high correlation with validated standards [15,16].

Elderly patients (>65 years) constitute a large and growing fraction of HNSCC patients due to demographic changes [17,18]. They are more vulnerable to treatment-related toxicity and deteriorated oncologic outcomes that may be further exacerbated by preexisting age-related sarcopenia [19,20]. A small retrospective study of 85 elderly HNSCC patients found pretherapeutic CT-based sarcopenia and low muscle function to be associated with deteriorated OS in patients who were mainly treated with surgery [21]. In the present analysis, we aimed to analyze sarcopenia and its dynamics over the course of curative (chemo)radiotherapy as well as its influence on oncological outcomes, treatment-related toxicities and treatment adherence in elderly HNSCC patients.

## 2. Methods

A total of 292 elderly HNSCC patients (>65 years) undergoing (chemo)radiation between 2010 and 2019 at the Department of Radiation Oncology, University of Freiburg Medical Center, were included, and 280 were analyzed in this study. 12 patients were excluded due to missing pretherapeutic weight information. The study was approved in advance by the institutional ethical review committee (reference no. 551/18). The patient cohort and treatment characteristics have been described earlier; for a detailed summary of patient characteristics see Appendix A [20]. 

For baseline sarcopenia assessment, the planning CT scan carried out at our institution with thermoplastic mask immobilization performed on a Siemens Emotion (5 mm slices, peak kilovoltage: 110) from 2010 to 2013 and from 2013 to 2019 on a Philips Brilliance Big Bore (2 mm slices, peak kilovoltage: 120) was used. For follow-up evaluation, the first post-therapeutic diagnostic CT scan was analyzed at three months after completion of radiotherapy. If there were no medical contraindications, intravenous iodine contrast administration was applied for all CT scans.

The cross-sectional muscle area (cm^2^) at the level of the third cervical vertebra was contoured and quantified. Therefore, the slice showing both transverse processes and the spinous process was identified and exported anonymously to the analyzation software ImageJ (US National Institutes of Health, Bethesda, MD, USA). The perivertebral muscles and both sternocleidomastoid muscles were manually outlined separately (thin yellow line) and segmentations were approved by a radiation oncologist. After contouring of the muscles’ compartment semi-automatic thresholding of muscle-specific Hounsfield-Units (HU −29 to +150) was applied to the outlined contours to exclude intramuscular fat tissue. The numeric value of the remaining muscle area inside of all three contours was automatically obtained via ImageJ. The muscle areas of the perivertebral muscles and both sternocleidomastoid muscles were summed up, resulting in the cross-sectional muscle area at the third cervical vertebra (C3MA). If one sternocleidomastoid muscle was infiltrated by the primary tumor or lymph node metastases, it was left out and the contralateral muscle was counted twice instead (Figure 1).

Based on a validated method published by Swartz et al., the cross-sectional muscle area at the third lumbar vertebra (L3MA) was calculated for each patient using the following formula [15]:L3MA (cm^2^) = 27.304 + (1.363 × C3MA) − (0.671 × age) + (0.640 × weight) + (26.442 × sex ^1^) 

^1^ (1 for female; 2 for male).

Calculated L3MA was then normalized for stature by dividing through the square of the patient’s height, giving the muscle index at the third lumbar vertebra (L3MI). Post-therapeutic values (pL3MA and pL3MI) were calculated accordingly.

For subgroup formation, patients were dichotomously categorized as sarcopenic or non-sarcopenic based on sex-specific cut-off values. The cut-off values were adopted from a publication by Derstine et al. [21] reporting muscle indices of a young and healthy reference cohort. Following the suggestion of the European working group on sarcopenia in older adults [5], the cut-off values were defined as two standard deviations below the mean of the afore mentioned reference cohort, amounting to 92.2 cm^2^ or 144.3 cm^2^ for L3MA in female and male patients and 34.3 cm^2^/m^2^ or 45.5 cm^2^/m^2^ for L3MI, respectively.

To analyze muscle mass as a continuous variable for both male and female patients together, each individual value for calculated L3MA and L3MI was standardized sex-specifically (i.e., L3MA_stand_ = (L3MA − mean_male_)/s.d._male_) with mean and standard deviations again taken from the reference cohort of Derstine et al. [22].

For survival analyses, Kaplan–Meier curves and uni- and multivariate Cox proportional hazard models were calculated. Overall survival (OS) was calculated from the last day of radiotherapy to death from any cause. Progression-free survival (PFS) was defined as the time from treatment completion to any progression of disease or death. Locoregional control (LRC) was equally calculated from completion of treatment until detection of metastases of the cervical lymph nodes or progression of the primary tumor. Survival data not available in the clinical record were obtained from the resident register of the federal state authorities of Baden-Württemberg. Subgroup comparisons between sarcopenic and non-sarcopenic patients were tested with Fisher’s exact tests for binominal comparisons and Mann–Whitney U tests for ordinal variables. Continuous variables were correlated with the calculated values for L3MA or L3MI by calculating Spearman’s rank correlation coefficients. All statistical data were calculated with SPSS statistics software version 27 (IBM Corp., Armonk, NY, USA). A *p*-value < 0.05 was considered statistically significant.

Toxicity was graded following CTCAE version 5. Chronic toxicity was defined as symptoms remaining or occurring at least 90 days after completion of radiotherapy. Chemotherapy was defined as completed for all patients receiving a cumulative dose of ≥200 mg/m^2^ cisplatin or  ≥450 mg/m^2^ carboplatin. Staging of HNSCCs was based on the 7th Edition of the UICC TNM classification. For calculation of the Charlson Comorbidity Index the current HNSCC was not excluded [23].

## 3. Results

### 3.1. Prevalence and Distribution of Pretherapeutic Sarcopenia in Elderly HNSCC Patients

When classified dichotomously based on literature-derived cut-off values for L3MA, more than half of our analyzed elderly HNSCC cohort (*n* = 155, 55.3%) was found to be sarcopenic. The rate was higher in men (57.6% vs. 50.0%). Median values for calculated L3MA and L3MI were 129.4 cm^2^ and 42.7 cm^2^/m^2^. When the individual values were standardized with a healthy reference cohort, our cohort exhibited a generally very low muscle mass with a median of 2.1 and 1.9 standard deviations below the average for L3MA and L3MI of the reference cohort, respectively (Figure 2) [22].

The percentage of locally advanced cancers (T3/4) was significantly higher in sarcopenic patients with 69.7% vs. 55.2% when dichotomized for L3MA (*p* = 0.001) and 68.1% vs. 58.1% for L3MI (*p* = 0.009). The mean Charlson Comorbidity Index was significantly higher in sarcopenic patients with 4.77 vs. 4.22 points (*p* = 0.015) for L3MA and 4.76 vs. 4.27 points (*p* = 0.032) for L3MI. No significant difference was observed for nodal involvement, smoking history or HPV status. Patients treated with primary (chemo)radiation showed a trend towards reduced skeletal muscle mass as compared to those treated postoperatively with 60.1% vs. 46.4% of sarcopenic patients for L3MA (*p* = 0.032) and 55.7% vs. 43.3% for L3MI (*p* = 0.059) (Table 1).

Continuous values of the calculated L3MA and L3MI correlated weakly but significantly with pretherapeutic hemoglobin levels (ρ = 0.280, *p* < 0.001 and ρ = 0.234, *p* < 0.001) and inversely patient age (ρ = −0.290, *p* < 0.001 and ρ = −0.198, *p* < 0.001). There was a strong correlation with pretherapeutic body weight (ρ = 0.702, *p* < 0.001 and ρ = 0.563, *p* < 0.001) The correlation with Karnofsky performance status was very weak (ρ = 0.156, *p* = 0.009 and ρ = 0.122, *p* = 0.041) as well as the inverse correlation with the Charlson Comorbidity Index (ρ = −0.197, *p* = 0.001 and ρ = −0.126, *p* = 0.035) (Table 2). 

### 3.2. Pretherapeutic Sarcopenia Is a Prognostic Marker in Elderly HNSCC Patients Undergoing (Chemo)Radiotherapy

Patients classified as sarcopenic after dichotomization for the calculated L3MA had significantly deteriorated overall survival with a median of 23 vs. 91 month and a 2-year survival rate of 49.6% vs. 71.1% (*p* = 0.002), compared to those not deemed sarcopenic. When classified based on the calculated L3MI, overall survival was found similarly deteriorated for sarcopenic patients with a median survival of 26 vs. 47 month and 2-year survival rates of 51.0% vs. 68.0%, respectively (*p* = 0.017) (Figure 3).

Locoregional control did not differ significantly between sarcopenic and non-sarcopenic patients neither after dichotomization for L3MA nor for L3MI with 2-year LRC of 71.2% vs. 78.5% (*p* = 0.378) and 71.5% vs. 77.9% (*p* = 0.732), respectively.

In a univariate Cox regression model, standardized continuous values of the calculated L3MA and L3MI were both significant factors of deteriorated overall survival with hazard ratios of 1.56 (CI 1.19–2.04, *p* < 0.001) and 1.33 (CI 1.03–1.72, *p* = 0.026). When dichotomized for L3MA and L3MI, both were associated with reduced survival with hazard ratios of 1.79 (CI 1.22–2.60, *p* = 0.003) and 1.55 (CI 1.08–2.24, *p* = 0.017).

In a multivariate analysis, including other factors known to be significant in this cohort from our previous analysis such as age, smoking status and Karnofsky Index, dichotomous L3MA remained statistically significant with a hazard ratio of 1.64 (CI 1.07–2.52, *p* = 0.023) (Table 3) [20].

Patients classified sarcopenic based on the calculated values for L3MI had significantly higher acute toxicities with 20.8%, 69.4% and 9.7% experiencing a maximum CTCAE grade 2, 3 and 4 toxicities, compared to 30.9%, 66.2% and 3.0% for non-sarcopenic patients (*p* = 0.011). When dichotomized for calculated values for L3MA, the difference was not statistically significant with 23.9/66.5%/9.7% vs. 28.0%/69.6%/2.4% (*p* = 0.103). The prevalence of chronic grade 3 or 4 toxicities was significantly higher in patients with pretherapeutic sarcopenia with 24.8 % vs. 11.8% (*p* = 0.022) dichotomized for L3MA, in contrast to an equal distribution for sarcopenic and non-sarcopenic patients categorized for L3MI with 20.3% vs. 17.0% (*p* = 0.514). Chronic maximum toxicity was not different for pretherapeutic sarcopenic and non-sarcopenic patients (Table 4).

Patients deemed sarcopenic (based on calculated values of L3MA) were significantly more likely to discontinue radiotherapy with 80.0% vs. 90.4% of patients receiving the scheduled radiotherapy dose (*p* = 0.040). A similar trend was seen for calculated L3MI sarcopenic patients with 80.6% vs. 89.0% completion rates (*p* = 0.093). There was no significant difference in chemotherapy completion rate with 58.1% vs. 62.4% (*p* = 0.540) and 57.6% vs. 62.5% (*p* = 0.464) for dichotomization analog L3MA and L3MI (Table 5). 

### 3.3. Elderly HNSCC Patients Show Minimal Post-Therapeutic Changes in Sarcopenia Prevalence and Distribution

Complete imaging-based post-therapeutic follow-up data was available for 109 patients. The median time from the beginning of radiotherapy to the follow-up CT imaging and the acquisition of follow-up weight was 127 days (4 month) and 125 days (4 month), respectively. Median loss of calculated skeletal muscle area at L3 (L3MA) and calculated skeletal muscle index at L3 (L3MI) were both 2.4% of the baseline value. Median weight loss during the course of therapy was 6.8% of patients’ body weight, which was significantly higher than loss of skeletal muscle indices (*p* < 0.001) (Figure 2).

When classified dichotomously analog literature-derived cut-off values for L3MA at the first evaluable follow-up, 67.9% of patients (*n* = 74) were sarcopenic. Approximately a third (*n* = 16, 32.6%) of former non-sarcopenic patients were now classified as sarcopenic, whereas 97% of former sarcopenic patients remained sarcopenic. When classified for L3MI, 60.6% of patients (*n* = 66) were sarcopenic at the first follow-up. Here, 29.6% (*n* = 16) of former non-sarcopenic patients were now classified as sarcopenic, whereas 91% of former sarcopenic patients remained sarcopenic (Figure 2).

### 3.4. Post-Therapeutic Sarcopenia Has No Prognostic Role in Elderly HNSCC Patients Undergoing (Chemo)Radiotherapy

Post-therapeutic sarcopenia was not associated with reduced overall survival with a median of 28 vs. 24 months (log-rank, *p* = 0.485) for sarcopenic and non-sarcopenic patients after dichotomization for pL3MA and 37 vs. 24 month (log-rank *p* = 0.466) for pL3MI, respectively (Figure 4). In a Cox regression model, neither pL3MA nor pL3MI significantly influenced OS. Neither did post-therapeutic sarcopenia significantly affect LRC. 

In a subgroup analysis, the particular group of patients changing from non-sarcopenic to sarcopenic over the course of therapy did not show a significantly different OS compared to the whole cohort (*p* = 0.519 for calculated L3MA and *p* = 0.632 for calculated L3MI) (Figure 4). Upon univariate Cox analysis, neither percental weight loss nor percental loss of calculated L3MA/L3MI showed a significant influence on OS with hazard ratios of 1.0 (CI 0.99–1.03; *p* = 0.395) and 0.99 (CI 0.95–1.02, *p* = 0.489).

The grade of chronic maximum toxicity (*p* = 0.351 for L3MA and *p* = 0.160 for L3MI) or the percentage of patients with grade 3/4 chronic toxicities (25.0% vs. 23.1%, *p* = 0.539 for L3MA and 22% vs. 29%, *p* = 0.314 for L3MI) were not significantly higher in patients classified as sarcopenic post-therapeutically. Patients classified sarcopenic after therapy were not significantly more likely to have discontinued radiotherapy before the scheduled dose (Appendix A). 

## 4. Discussion

In our cohort, sarcopenia derived from skeletal muscle mass was common in elderly patients receiving curative (chemo)radiotherapy for HNSCC. Pretherapeutic sarcopenia was associated with significantly deteriorated OS but not with altered LRC. In addition, skeletal muscle mass correlated significantly but weakly with hemoglobin levels, and negatively with patient age. Patients classified as sarcopenic featured larger tumors and a higher burden of comorbidity. Over the course of therapy, sarcopenia was relatively stable compared to the observed weight loss in elderly HNSCC patients. Interestingly, neither weight loss nor the loss of skeletal muscle mass over the course of therapy nor post-therapeutic sarcopenia were associated with patient survival. 

The value of sarcopenia as a prognostic factor in HNSCC patients has been widely discussed. Chargi et al. found skeletal muscle mass-derived sarcopenia to be significantly associated with dose-limiting toxicities of cisplatin for concomitant chemoradiation [24]. Similarly, Ganju et al. described a significantly increased risk for treatment interruptions in sarcopenic HNSCC patients undergoing radiotherapy, and poor treatment tolerance in sarcopenic patients has also been reported for surgical approaches [25,26,27]. 

The prognostic value of sarcopenia on overall survival has been subject to two meta-analyses by Findlay et al. [14,28]. The first meta-analysis covering approximately one thousand patients showed a highly significant association of pre- and post-therapeutic sarcopenia with reduced overall survival in 7 studies with median patient ages of 57–66 years. Sarcopenia was classified on CT at the height of the third lumbar vertebra in all analyses [14]. A second meta-analysis with 3400 patients from 11 studies included sarcopenia defined at the level of the third cervical vertebra and mainly confirmed the previous results, yet patient age was not reported in the updated version [28]. 

Although multiple authors have described the association of advanced age and a higher prevalence of sarcopenia in HNSCC patients [29,30], sarcopenia dynamics and its prognostic role specifically for elderly HNSCC patients has rarely been studied. Chargi et al. reported an influence of sarcopenia in a small cohort of elderly HNSCC patients treated with curative intent, although the majority of patients in this publication did not undergo radiotherapy. With 48.2%, the rate of sarcopenic patients as defined by low skeletal muscle mass at the level of C3 and low muscle strength was comparable to our results, despite the somewhat older median age of 81 years. The reported high prognostic influence of pretherapeutic sarcopenia on OS paralleled our results, suggesting a similar prognostic role of sarcopenia for surgical and radiotherapeutic treatment approaches. Of note, cut-off values for defining sarcopenia in Chargi’s work were not sex-specific, and no data were provided on post-therapeutic sarcopenia.

In our cohort of elderly HNSCC patients, pre-treatment sarcopenia had a strong prognostic relevance compared to the post-treatment evaluation. In contrast, Ahern et al. reported reduced OS for patients with post-therapeutic sarcopenia but no influence on survival of pretherapeutic sarcopenia [29], hereby deviating from many publications reporting deteriorated OS in initially sarcopenic HNSCC patients [13,30]. In the meta-analysis by Findlay et al., two studies included pre- and post-therapeutic sarcopenia, and a highly significant prognostic influence was concluded for both. 

Our data emphasize the role of assessing pre-treatment sarcopenia in elderly HNSCC patients. Interestingly, in comparison to weight loss over the course of therapy, sarcopenia remained relatively stable and although being considerably more pronounced, weight loss during therapy seemed to be of less prognostic relevance, which has been reported from prospective data earlier [31].

The used methodology is crucial to the scientific evaluation of sarcopenia. According to the updated consensus of the European Working Group on Sarcopenia in Older People, sarcopenia can be determined by a many different indicators including radiologic measurements of muscle mass [5]. Although whole-body MRI is seen as the imaging gold standard, measurement on single transversal slices of cross-sectional imaging is widely used for the evaluation of skeletal muscle mass in oncology with CT scans showing equivalently accurate results [32,33,34,35]. Especially the skeletal muscle area at the level of the third lumbar vertebra (L3MA) has been validated against whole body MRI [11,36,37]. Cross-sectional muscle mass at the level of the third cervical vertebra (C3MA) reportedly shows a strong correlation with L3 [36,37]. In this respect, Swartz et al. showed a strong correlation between L3MA and C3MA, and proposed a multivariate prediction model including patients’ weight, age and sex, showing excellent correlation of L3MA with calculated L3MA from C3MA. This prediction rule was used in the present study. A recent study by Vangelov et al. questioned the use of C3MA as an evaluation tool of skeletal muscle mass in HNSCC patients [38]. Although the authors found a good correlation of measured L3MA and calculated L3MA from C3MA, systemic biases were reported with underestimation of L3MA for low values of C3MA and overestimation for high C3MA values and therefore hampering classification of individual patients into sarcopenic and non-sarcopenic. Clinical data however seem to level these differences. A large meta-analysis by Wong et al. on the influence of CT-defined sarcopenia for OS in HNSCC patients found similar hazard ratios for C3MA and L3MA-derived sarcopenia measurements [13].

Our study faces the inherent limitations of a retrospective analysis. Unfortunately, some patients only received MRI as follow-up imaging. The difficulties in intermodal comparison, especially the intra- and inter-examination variations of MRI grey-scale values in contrast to the standardized Hounsfield units impaired the use of MRI data for follow-up analysis and therefore reduced the number of patients with follow-up sarcopenia information. To rule out possible selection biases, we compared the subgroups of patients with and without follow-up CT imaging. The mean values of pre-treatment sarcopenia were equal for both groups suggesting no such bias.

Despite these limitations, we could demonstrate the prognostic value of pretherapeutic sarcopenia in a large cohort of elderly HNSCC patients, that are especially prone to age and disease-related loss of skeletal muscle mass. In contrast, post-therapeutic sarcopenia and peritherapeutic weight loss were of insignificant prognostic value in this cohort. This highlights the importance of careful patient selection where sarcopenia should be a factor worth considering in the pretherapeutic decision-making process to identify vulnerable patients that experience unfavorable outcomes despite aggressive therapy [39,40,41]. Further, assessment of pretherapeutic sarcopenia could be used for tailored supportive interventions such as physical therapy or nutrition counseling. Our data suggest that these interventions should start as early as possible, i.e., at diagnosis. This aspect should be further evaluated prospectively.

## Figures and Tables

**Figure 1 cancers-14-05536-f001:**
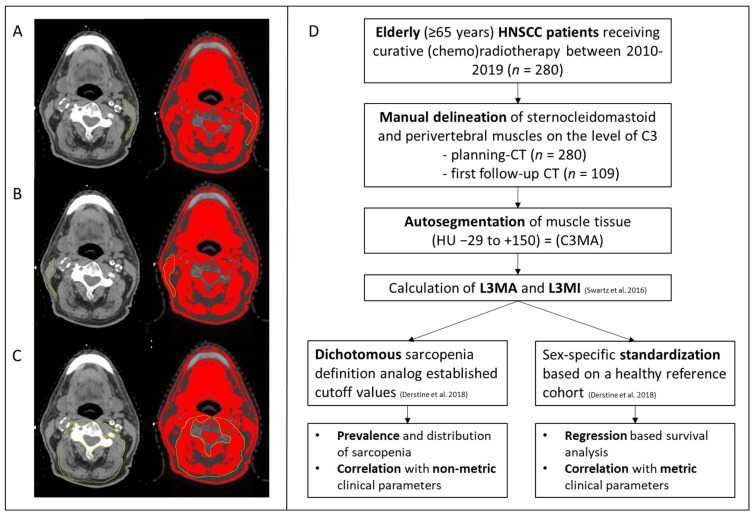
Methodology of skeletal muscle mass assessment for sarcopenia determination in HNSCC patients. Manual delineation (thin yellow contour) of the left sternocleidomastoid muscle (**A**), right sternocleidomastoid muscle (**B**) and paravertebral muscles (**C**) at the level of C3 with Hounsfield-Unit guided autosegmentation of muscle tissue (red area). (**D**)Flow diagram of data acquisition and data processing. HNSCC, head and neck squamous cell carcinoma; C3, third cervical vertebra; CT, computed tomography; HU, Hounsfiled-Units; C3MA, cross-sectional skeletal muscle area at the third cervical vertebra; L3MA, cross-sectional skeletal muscle area at the third lumbar vertebra; L3MI, skeletal muscle index at the third lumbar vertebra [15,22].

**Figure 2 cancers-14-05536-f002:**
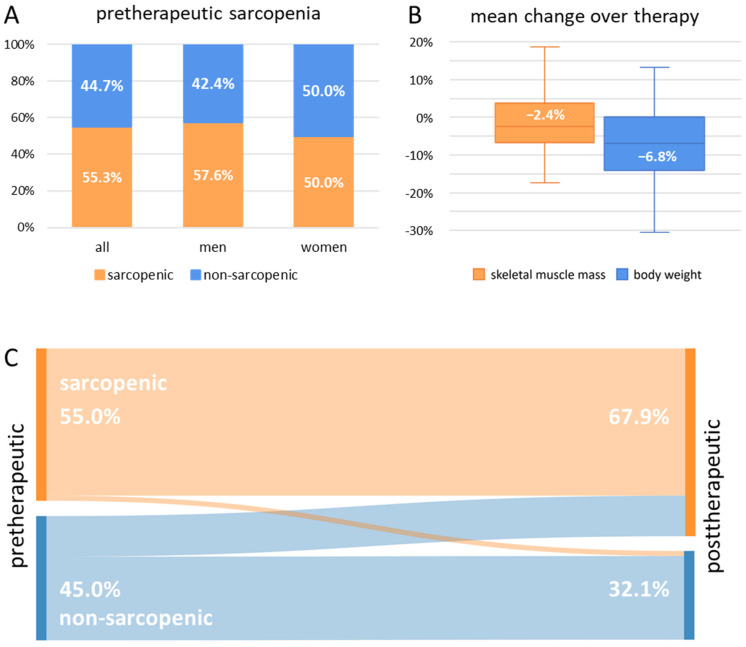
(**A**): Fraction of pretherapeutic sarcopenic and non-sarcopenic elderly HNSCC patients dichotomously classified for the calculated L3MA separately shown for the whole cohort, male and female patients. (**B**): Boxplot showing mean change of patients’ calculated skeletal muscle index L3MA and patients’ body weight over the course of radiotherapy. (**C**): Flow diagram of the changes in the fraction of sarcopenic and non-sarcopenic patients over the course of radiotherapy, classified dichotomously for L3MA, given for all patients with follow-up data. HNSCC, head and neck squamous cell carcinoma; L3MA, cross-sectional skeletal muscle area at the third lumbar vertebra.

**Figure 3 cancers-14-05536-f003:**
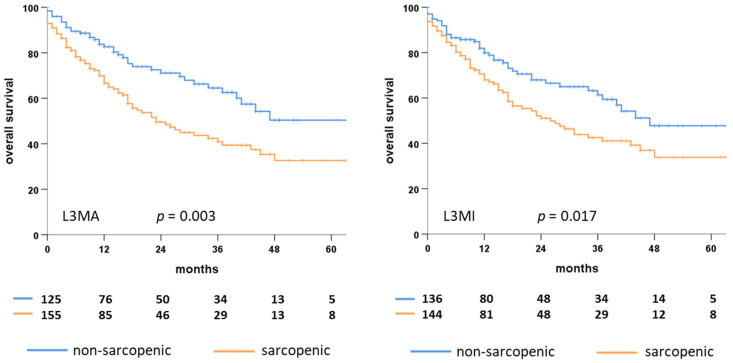
Kaplan Meier plots showing overall survival of pretherapeutic sarcopenic (orange) and non-sarcopenic (blue) elderly HNSCC patients classified for L3MA (left plot) and L3MI (right plot). L3MA, calculated cross-sectional skeletal muscle area at the third lumbar vertebra; L3MI, calculated skeletal muscle index at the third lumbar vertebra; *p*, *p*-value; HNSCC, head and neck squamous cell carcinoma.

**Figure 4 cancers-14-05536-f004:**
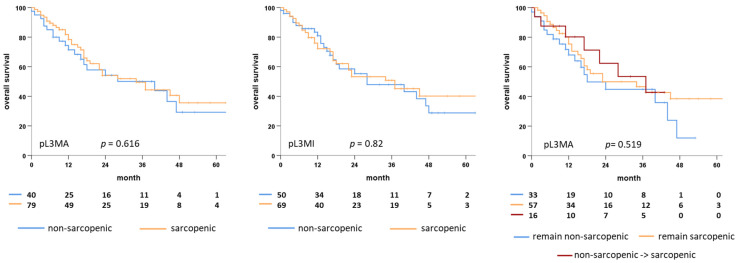
Kaplan Meier plots showing overall survival of post-therapeutic sarcopenic (orange) and non-sarcopenic (blue) elderly HNSCC patients classified for pL3MA (left plot) and pL3MI (middle plot). Right Kaplan Meier plot shows overall survival of three subgroups: blue (patients remaining non-sarcopenic pre- and post-therapeutically) orange (patients remaining sarcopenic pre- and post-therapeutically) and red (pretherapeutic non-sarcopenic patients developing sarcopenia over the course of therapy), all classified for L3MA and pL3MA. pL3MA, calculated post-therapeutic cross-sectional skeletal muscle area at the third lumbar vertebra; pL3MI, calculated post-therapeutic skeletal muscle index at the third lumbar vertebra.

**Table 1 cancers-14-05536-t001:** Comparison of clinico-pathological parameters for pretherapeutic sarcopenic and non-sarcopenic elderly HNSCC patients undergoing curative (chemo)radiotherapy. *p* values given for group comparison with Mann–Whitney U tests and Fisher’s exact tests. L3MA, calculated cross-sectional skeletal muscle area at the third lumbar vertebra; L3MI, calculated skeletal muscle index at the third lumbar vertebra; *n*, number of patients; CUP, cancer of unknown primary; HPV, human papillomavirus; Karnofsky, Karnofsky Performance Status; CCI, Charlson Comorbidity Index.

		L3MA					L3MI				
*n* = 280		Non-Sarcopenic		Sarcopenic			Non-Sarcopenic		Sarcopenic		
		(*n* = 125)		(*n* = 155)			(*n* = 136)		(*n* = 144)		
**T-stage**	**T1**	22	17.6%	15	9.7%	*p* = 0.001	21	15.4%	16	11.1%	*p* = 0.009
	**T2**	33	26.4%	30	19.4%		35	25.7%	28	19.4%	
	**T3**	35	28.0%	38	24.5%		40	29.4%	33	22.9%	
	**T4**	34	27.2%	70	45.2%		39	28.7%	65	45.1%	
	**CUP**	2	1.6%	1	0.6%		2	1.5%	1	0.7%	
**N-stage**	**N0**	47	37.6%	59	38%	*p* = 0.763	55	40.4%	51	35.4%	*p* = 0.645
	**N1**	18	14.4%	24	15.5%		17	12.5%	25	17.4%	
	**N2**	53	42.4%	66	42.6%		58	42.6%	61	42.4%	
	**N3**	7	5.6%	6	3.9%		6	4.4%	7	4.9%	
**HPV**	**+**	26	20.8%	22	14.2%	*p* = 0.574	24	17.6%	24	16.7%	*p* = 0.711
	**-**	33	26.4%	36	23.2%		37	27.2%	32	22.2%	
	**n/a**	66	52.8%	97	62.6%		75	55.1%	88	61.1%	
**smoker**	**yes**	69	55.2%	92	59.4%	*p* = 0.157	79	58.1%	82	56.9%	*p* = 0.574
	**no**	39	31.2%	34	21.9%		39	28.7%	34	23.6%	
	**n/a**	17	13.6%	29	18.7%		18	13.2	28	19.4%	
**Karnofsky**	**100–90**	73	58.4%	80	51.6%	*p* = 0.093	76	55.9%	77	53.5%	*p* = 0.371
	**80–70**	48	38.4%	56	36.1%		54	39.7%	50	34.7%	
	**60–50**	4	3.2%	18	11.6%		5	3.7%	17	11.8%	
	**40–30**	0	0.0%	1	0.7%		1	0.7%	0	0.0%	
	**n/a**	0	0.0%	0	0%		0	0.0%	0	0.0%	
**CCI**	**mean**	4.22	-	4.77	-	*p* = 0.015	4.27	-	4.76	-	*p* = 0.032

**Table 2 cancers-14-05536-t002:** Correlation of clinical variables with standardized pretherapeutic skeletal muscle indices L3MA and L3MI. Spearman’s rank correlation coefficients and corresponding *p* values are given. L3MA, calculated cross-sectional skeletal muscle area at the third lumbar vertebra; L3MI, calculated skeletal muscle index at the third lumbar vertebra; Hb, hemoglobin; RT, radiotherapy; CRP, C-reactive protein.

Spearman Correlation for Continuous Sex-Standardized Sarcopenia Parameters	L3MA		L3MI	
	Corr. Coeff.	*p*-Value	Corr. Coeff.	*p*-Value
**Hb** (pre RT)	0.280	<0.001	0.234	<0.001
**albumin** (pre RT)	0.161	0.046	0.127	0.118
**CRP** (pre RT)	−0.150	0.016	–0.070	0.265
**Creatinin** (pre RT)	0.072	0.246	0.104	0.094
**age**	−0.290	<0.001	−0.198	0.001
**Karnofsky**	0.156	0.009	0.122	0.041
**body weight** (pre RT)	0.702	<0.001	0.563	<0.001
**Charlson Comorbidity Index**	−0.197	0.001	−0.126	0.035
**acute max. toxicity**	−0.111	0.063	−0.147	0.014
**chronic max. toxicity**	−0.001	0.992	0.053	0.435

**Table 3 cancers-14-05536-t003:** Univariate Cox regression models of sarcopenia parameters for overall survival. Dichotomous classifications compare non-sarcopenic patients (i.e., high values, i.e., >cut-off) vs. sarcopenic patients (i.e., low values, i.e., ≤cut-off). HR and CI values of the continuous variables have been inverted to facilitate comparability with the dichotomous classification. The strongest skeletal muscle index “L3MA dichotomous” was included in a multivariate regression model with known factors of deteriorated overall survival. L3MA, calculated cross-sectional skeletal muscle area at the third lumbar vertebra; L3MI, calculated skeletal muscle index at the third lumbar vertebra; HR = hazard ratio, OS = overall survival, CI = confidence interval.

Univariate	HR for OS	CI 95%	*p*-Value
LM3A dichotomous	1.786	1.22–2.60	0.003
LM3I dichotomous	1.553	1.08–2.24	0.017
LM3A continuous	1.562	1.19–2.04	0.001
LM3I continuous	1.331	1.03–1.72	0.026
**Multivariate**	**HR for OS**	**CI 95%**	***p*-value**
LM3A dichotomous	1.64	1.07–2.52	0.023
age (65–74 vs. ≥75 yrs)	1.44	0.94–2.20	0.091
smoking	1.88	1.14–3.10	0.013
Karnofsky (1 unit increment)	0.965	0.95–0.98	<0.001

**Table 4 cancers-14-05536-t004:** Comparison of treatment-related toxicities following the CTCAE v.5 for pretherapeutic sarcopenic and non-sarcopenic elderly HNSCC patients undergoing curative (chemo)radiotherapy. *p* values given for group comparison with Mann–Whitney U tests and Fisher’s exact tests. L3MA, calculated cross-sectional skeletal muscle area at the third lumbar vertebra; L3MI, calculated skeletal muscle index at the third lumbar vertebra; *n*, number of patients.

		L3MA					L3MI				
*n* = 280		Non-Sarcopenic		Sarcopenic			Non-Sarcopenic		Sarcopenic		
		(*n* = 125)		(*n* = 155)			(*n* = 136)		(*n* = 144)		
**Acute max. toxicity**	**2**	35	28.0%	37	23.9%	*p* = 0.103	42	30.9%	30	20.8%	*p* = 0.011
	**3**	87	69.6%	103	66.5%		90	66.2%	100	69.4%	
	**4**	3	2.4%	15	9.7%		4	3.0%	14	9.7%	
	**n/a**	0	0.0%	0	0.0%		0	0.0%	0	0.0%	
**Acute max. toxicity** exclusive hematological	**0**	0	0.0%	9	5.8%	*p* = 0.368	1	0.7%	8	5.6%	*p* = 0.708
	**1**	7	5.6%	11	7.2%		8	5.9%	10	6.9%	
	**2**	42	33.6%	45	29%		50	36.8%	37	25.7%	
	**3**	74	59.2%	87	56.1%		75	55.1%	86	59.7%	
	**4**	2	1.6%	3	1.9%		2	1.5%	3	2.1%	
	**n/a**	0	0.0%	0	0.0%		0	0.0%	0	0.0%	
**Chronic toxicity** ***n* = 218**		***n* = 93**		***n* = 125**			***n* = 100**		***n* = 118**		
**Chronic max. toxicity**	**0**	7	7.5%	22	18.9%	*p* = 0.962	8	8%	21	17.8%	*p* = 0.584
	**1**	13	14%	17	13.6%		16	16%	14	11.9%	
	**2**	61	65.6%	55	44%		58	58%	58	47.2%	
	**3**	11	11.8%	29	23.2%		17	17%	23	19.5%	
	**4**	0	0.0%	1	0.8%		0	0.0%	1	0.9%	
**Chronic** **toxicity** cumulative	**1**	62	66.7%	75	60%	*p* = 0.315	65	65%	72	61%	*p* = 0.545
	**2**	69	74.1%	76	60.8%	*p* = 0.039	72	72%	73	61.9%	*p* = 0.115
	**3**	11	11.8%	30	24%	*p* = 0.023	17	17%	24	20.3%	*p* = 0.531
	**4**	0	0.0%	1	0.8%	*p* = 0.388	0	0.0%	1	0.9%	*p* = 0.357
**Chronic max.** **toxicity grade 3/4**		11	11.8%	30	24.8%	*p* = 0.022	17	17%	24	20.3%	*p* = 0.514

**Table 5 cancers-14-05536-t005:** Comparison of treatment characteristics for pretherapeutic sarcopenic and non-sarcopenic elderly HNSCC patients undergoing curative (chemo)radiotherapy. *p* values given for group comparison with Fisher’s exact tests. L3MA, calculated cross-sectional skeletal muscle area at the third lumbar vertebra; L3MI, calculated skeletal muscle index at the third lumbar vertebra; *n*, number of patients.

		L3MA					L3MI				
*n* = 280		Non-Sarcopenic		Sarcopenic			Non-Sarcopenic		Sarcopenic		
		(*n* = 125)		(*n* = 155)			(*n* = 136)		(*n* = 144)		
**chemotherapy** **planned**	**no**	47	37.6%	65	41.9%	*p* = 0.540	51	37.5%	61	42.4%	*p* = 0.464
	**yes**	78	62.4%	90	58.1%		85	62.5%	83	57.6%	
	**n/a**	0	0.0%	0	0.0%		0	0.0%	0	0.0%	
**chemotherapy completed**	**yes**	45	36%	59	64.4%	*p* = 0.339	47	34.6%	57	39.6%	*p* = 0.183
	**no**	19	15.2%	17	20.0%		21	15.4%	15	10.4%	
	**n/a**	61	48.8%	79	51.0%		68	50%	72	50%	
**RT completed**	**yes**	113	90.4%	124	80%	*p* = 0.040	121	89%	116	80.6%	*p* = 0.093
	**no**	12	9.6%	29	18.7%		15	11%	26	18.1%	
	**n/a**	0	0.0%	2	1.3%		0	0.0%	2	1.4%	

## Data Availability

The data presented in this study are available anonymized on request from the corresponding author. The data are not publicly available due to data privacy.

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
