# Peer review of "Sarcopenia as a Prognostic Marker in Elderly Head and Neck Squamous Cell Carcinoma Patients Undergoing (Chemo-)Radiation"

_cancers, 2022, doi:10.3390/cancers14225536_

Round 1

Reviewer 1 Report

Overall, the statistical methodology applied appears solid and the reporting and interprtetation of results is accurate.

I only suggest to fix the following specific issues:

- page 3, lines 112-115: please clarify how the cut-offs for the sarcopenic indices were calculated. I do not understand the meaning of the sentence "the cutoff values provided mean data and two standard deviations". In addition, for reproducibility and reference for future comparisons, you should report the values of these cut-offs.

Moreover, although the interpretation of the association between sarcopenia and OS is easier after dichotomzation of the indexes, it could be of interest to analyze this association considering the indexes as continous, for example by including them as covariates in the Cox model with a spline transformation. This could be useful also to judge the adequacy of the chosen cut-offs.

- page 7, lines 173-177. Please, do not rely on the p-values to judge the correlations. The p-value here only informs on whether the true correlation index is likely to be different from 0. This does not imply that the correlation is high or relevant. Values of rho<0.2 are generally regarded as indicator of a very weak correlation. The p-value depends on the sample size so if you have a sample large enough, even a negligible correlation (say rho=0.01) could be statistically significant.

- Figure 3 and figure 4. Please report the number of patients still at risk at various times (e.g. every 12 months) below each plot.

- Table 3. Please specifiy in the table that the reported HRs for dichotomous sarcopenic indices are comparing low values, i.e. < cutoff, vs high values, i.e. > cutoff (or else?). Put also the equal sign in the right place (<= vs > or < vs >= ?). Is the reported HR of age related to an increment of 1 year? Is the reported HR for Karnofsky related to an increment of 1 unit? Please specificy in the table.

Author Response

Overall, the statistical methodology applied appears solid and the reporting and interpretation of results is accurate.

- We thank the reviewer for his positive feedback and helpful suggestions.

I only suggest to fix the following specific issues:

page 3, lines 112-115: please clarify how the cut-offs for the sarcopenic indices were calculated. I do not understand the meaning of the sentence "the cutoff values provided mean data and two standard deviations". In addition, for reproducibility and reference for future comparisons, you should report the values of these cut-offs.

- We agree that the description of the cut-off calculation was lacking comprehensibility and specified the referring section accordingly. The sentence now reads “The cut-off values were adopted from a publication by Derstine et al. [21] reporting muscle indices of a young and healthy reference cohort. Following the suggestion of the European working group on sarcopenia in older adults [5], the cut-off values were defined as two standard deviations below the mean of the aforementioned reference cohort, amounting to 92.2cm2 or 144.3cm2 for L3MA in female and male patients and 34.3cm2/m2 or 45.5 cm2/m for L3MI, respectively.”

Moreover, although the interpretation of the association between sarcopenia and OS is easier after dichotomzation of the indexes, it could be of interest to analyze this association considering the indexes as continuous, for example by including them as covariates in the Cox model with a spline transformation. This could be useful also to judge the adequacy of the chosen cut-offs.

- We agree that beyond dichotomized values, analysis of continuous muscle indices is of high interest. Table 3 shows Cox regression models for the influence of L3MA and L3MI on overall survival both as dichotomized and continuous variables, showing significantly deteriorated survival for all of them. In addition, the strongest predictor in the univariate model was then included in a multivariate model with other known factors of reduced survival.

page 7, lines 173-177. Please, do not rely on the p-values to judge the correlations. The p-value here only informs on whether the true correlation index is likely to be different from 0. This does not imply that the correlation is high or relevant. Values of rho<0.2 are generally regarded as indicator of a very weak correlation. The p-value depends on the sample size so if you have a sample large enough, even a negligible correlation (say rho=0.01) could be statistically significant.

- We thank the reviewer for his helpful comment and concur that the evaluation of correlations requires rho-values. Therefore, the p-values in the abstract were replaced with the respective rho-values. In the manuscript, all mentioned correlations were textually classified following the suggestions from Cohen and Evans.

Figure 3 and figure 4. Please report the number of patients still at risk at various times (e.g. every 12 months) below each plot.

- The number of patients-at-risk was added to the respective figures.

Table 3. Please specifiy in the table that the reported HRs for dichotomous sarcopenic indices are comparing low values, i.e. < cutoff, vs high values, i.e. > cutoff (or else?). Put also the equal sign in the right place (<= vs > or < vs >= ?). Is the reported HR of age related to an increment of 1 year? Is the reported HR for Karnofsky related to an increment of 1 unit? Please specificy in the table.

- The reported HRs for the dichotomous classifications compare non-sarcopenic patients (“0”, i.e. high values, i.e. > cut-off) vs sarcopenic patients (“1”, i.e. low values, i.e. £ cut-off) resulting in HRs >1 for overall survival, indicating deteriorated survival for sarcopenic patients. This has been specified in the table caption.

Since skeletal muscle indices analyzed as continuous variables would result in HRs <0 (high values leading to improved survival), the HRs and CIs of the continuous analysis were inverted to facilitate comparability with the dichotomous analysis. This is now stated in the table caption.

The reported HR for age compared patients aged 65-74 years with those over ³75 years. The reported HR for the Karnofsky Performance Score relates to an increment of 1 unit. Both details have been now specified in the table.

Reviewer 2 Report

Cancers

Title: Sarcopenia as a prognostic marker in elderly head neck squamous cell carcinoma patients undergoing (chemo)-radiation

In their manuscript, entiteled „Sarcopenia as a prognostic marker in elderly head neck squamous cell carcinoma patients undergoing (chemo)-radiationthe authors have analyzed sarcopenia and sacropenia-dynamics in HNSCC patients undergoing (chemo)radiation.

The strength of this study is that, the results are relevant in the daily clinical workflow. The study design is plausible, the manuscript has a clear structure and the included patient population has a reasonable sample size.

I highly recommend some revisions as outlined in the following:

-       Introduction:

Please adjust the definition of cancer cachexia (line 49) as stated by Fearon et al 2011 (PMID: 21296615)

-       Material and Methods/Results:

In the material and methods section the authors described that L3MA/L3MI were not assessed directly by measuring the lumbar muscles but were calculated. This circumstance has to be highlighted in the whole manuscript by not writing about L3MA/L3Mi but about “calculated L3MA/L3MI”. Otherwise this is really misleading!

-       Material and Methods:

Please mention who did the muscle segmentation, was it a radiation-oncologist? In the list of the authors 3 different radio-oncological centres are mentioned. Did they also provide data? If yes, was there a central review of data analysis?

-       Figure 1:

All images in this figure have to be depicted with A/B/C and there have to be linked descriptions in the figure caption. 

Also it should clearly state what structures are marked in this images because besides muscles also vascular structures, facias, ect. are marked, thus I wonder if these structures were included intentional had an impact on the results the authors are representing or unintentional (in this case I would recommend to include a radiologist in the data analysis). 

-       Figures/Tables:

All tables and figures should include an abbreviation list in their figure/table capture.

Author Response

In their manuscript, entiteled „Sarcopenia as a prognostic marker in elderly head neck squamous cell carcinoma patients undergoing (chemo)-radiation” the authors have analyzed sarcopenia and sacropenia-dynamics in HNSCC patients undergoing (chemo)radiation.

The strength of this study is that, the results are relevant in the daily clinical workflow. The study design is plausible, the manuscript has a clear structure and the included patient population has a reasonable sample size.

- We thank the reviewer for the appraisal and the detailed feedback.

I highly recommend some revisions as outlined in the following:

Introduction:

Please adjust the definition of cancer cachexia (line 49) as stated by Fearon et al 2011 (PMID: 21296615)

- We thank the reviewer for indication of the renewed definition of cancer cachexia and adjusted the respective section in the introduction of the manuscript.

Material and Methods/Results:

In the material and methods section the authors described that L3MA/L3MI were not assessed directly by measuring the lumbar muscles but were calculated. This circumstance has to be highlighted in the whole manuscript by not writing about L3MA/L3Mi but about “calculated L3MA/L3MI”. Otherwise this is really misleading!

- The calculation of L3MA/L3MI followed an established approach as described in the methods section of the abstract and the manuscript. To highlight this procedure, L3MA and L3MI was stated as “calculated” in all figures and tables as well as at the first mention in each paragraph throughout the manuscript.

Material and Methods:

Please mention who did the muscle segmentation, was it a radiation-oncologist? In the list of the authors 3 different radio-oncological centres are mentioned. Did they also provide data? If yes, was there a central review of data analysis?

- The muscle segmentation was approved by a radiation oncologist. This was specified in the method section.

We apologize for the confusing affiliations. All patients were treated at the University of Freiburg Medical Center; thus this is a large single-center analysis. At the time of data acquisition and analysis, all authors were employed at this institution. During manuscript preparation, two authors, EH and NHN, left the University Hospital of Freiburg and are now employed by the University Hospital of Munich (LMU) and the University Hospital of Leipzig, respectively.

Figure 1:

All images in this figure have to be depicted with A/B/C and there have to be linked descriptions in the figure caption.

- The figures were modified according to the reviewer’s suggestion.

Also it should clearly state what structures are marked in this images because besides muscles also vascular structures, facias, ect. are marked, thus I wonder if these structures were included intentional had an impact on the results the authors are representing or unintentional (in this case I would recommend to include a radiologist in the data analysis).

- We apologize for the unclear description of the methodology. The muscle compartment of both sternocleidomastoid (A, B) and the perevetebral muscles (C) were manually contoured separately (thin yellow line). Then autosegmentation of muscle-specific Hounsfiled units was performed to exclude intramuscular fat (red area). Only the red area inside the muscle compartment (yellow line) was used for the calculation and analysis. We revised the respective part of the method section accordingly.

The described method has been used and published by different authors to assess skeletal muscle mass on CT scans of head and neck cancer patients [15;16].

Figures/Tables:

All tables and figures should include an abbreviation list in their figure/table capture.

- As suggested by the reviewer, abbreviation lists were added to all figure/table captions.

Round 2

Reviewer 2 Report

The authors have satisfactorily taken into account all my suggestions